# Genome-Wide Identification and Analysis of Collar Region-Preferential Genes in Rice

**DOI:** 10.3390/plants12162959

**Published:** 2023-08-16

**Authors:** Xu Jiang, Woo-Jong Hong, Su-Kyoung Lee, Ki-Hong Jung

**Affiliations:** 1Graduate School of Green-Bio Science and Crop Biotech Institute, Kyung Hee University, Yongin 17104, Republic of Korea; kangwuk97@khu.ac.kr (X.J.); aromy71@khu.ac.kr (S.-K.L.); 2Department of Smart Farm Science, Kyung Hee University, Yongin 17104, Republic of Korea; hwj0602@khu.ac.kr

**Keywords:** collar region, transcriptome analysis, leaf architecture, rice, *lg1* mutant

## Abstract

The collar region plays a crucial role in leaf angle formation and plant architecture, which is important for improving crop yield given the challenges of diminishing arable land and changing environmental conditions. To determine collar region-preferential genes (CRPGs) affecting plant architecture and crop yield, we conducted genome-wide transcriptomic analysis. By integrating our RNA sequencing data with public rice anatomical expression data, we identified 657 CRPGs. Verification involved testing six randomly selected CRPGs, all of which exhibited collar-preferential expression. The functional significance of CRPGs was assessed via Gene Ontology enrichment analysis, utilizing MapMan and KEGG, and literature analysis provided additional information for characterized CRPGs. Our findings revealed links between manipulating leaf angle and phytohormone-related pathways and stress responses. Moreover, based on the CRPGs, five transcription factors downstream of the *liguleless 1 (LG1)* gene were identified. Overall, the identified CRPGs provide potential targets for further research and breeding applications aimed at improving crop productivity by manipulating leaf architecture.

## 1. Introduction

Improving crop yield is an urgent priority owing to complex environmental issues and growing populations [1,2,3]. To meet the global demand for food, efforts to enhance productivity in rice (*Oryza sativa* L.), one of the world’s most important crops, have focused on manipulating leaf architecture [4]. Rice leaf architecture consists of two distinguishable organs: the vibrant green leaf area and the degenerated white-collar region (Figure 1a). The former facilitates photosynthesis and respiration, whereas the latter shapes the crucial morphological structure known as leaf angle (LA) [5]. Accumulated evidence suggests that the erect leaf phenotype, a key trait in plant leaf architecture, is beneficial for enhancing light capture and photosynthesis efficiency under dense planting, thereby increasing yield [4,6,7]. On the other hand, LA plasticity provides plant flexibility in adapting to rapidly changing environmental conditions [8,9]. Therefore, understanding the collar region comprehensively may contribute to fine-tuning LA for improved breeding applications.

Sequential sectioning has been used to investigate morphological changes and the cytological basis of the collar region during its development [10,11]. The cell type divergence and cytological transitions of sclerenchyma and aerenchyma from parenchyma cells and the asymmetric cell constitutions and elongation at the antithetical sides of the cell resulted in LA formation. Dynamic changes in LA are influenced by associated processes, including cell division, cell expansion, and cell wall compositional change [12,13,14]. For instance, overexpression of *ILI1* promotes leaf inclination through cell elongation on the adaxial side [15]. Moreover, extensive studies revealed the involvement of multiple factors like various phytohormone-associated genes and transcription factors (TFs) in the leaf angle determination. Intricate phytohormone regulatory networks are an important aspect of collar region development and LA regulation. *FISH BONE (FIB)* acts as an auxin biosynthesis gene and decreases indole-3-acetic acid content, and thus, enlarged LA was caused by the mutation of *FIB* [16]. Also, the overexpression of auxin signaling genes, including *OsGH3-2* [17], *OsIAA1* [18], and *OsARF19* [19], indicated that auxin negatively affects leaf inclination. Conversely, the promoting effect of brassinosteroids in leaf angle enlargement and its crosstalk control with other phytohormones, such as auxin, gibberellin, and abscisic acid, have been revealed by functional studies [20,21,22,23,24]. Among many TFs, *LIGULELESSs* play dominant roles in initiating collar region organogenesis, which is fundamental to LA formation. *OsLG2/2L* gene-edited plants exhibit localization perturbations in the boundary between blade and sheath, which further disrupt *OsLG1*-mediated collar differentiation [25,26]. This organogenesis initiation triggers well-ordered cytological changes to occur, leading to leaf bending.

Via systemic dissection with a focus on collar region development, many stage-specific genes have been identified, and they have been proposed to be potential targets for LA manipulation [10,11]. It is helpful for illustrating the complex regulatory mechanisms of collar region development. However, despite a few examples of success in yield enhancement by modulating these genes, the pleiotropic expression still hinders their practical engineering and application. This multi-functional phenomenon of a single gene is often accompanied by negative effects on plant growth, making it difficult to obtain the ideal leaf phenotype without affecting other developmental processes. For example, although RNA interference mutants of *OsBUL1* exhibit the erect leaf phenotype, their grain size is significantly reduced, adversely affecting final productivity [27]. To mitigate these adverse impacts, it is important to identify tissue-preferential genes. Modifying these genes may achieve the desired phenotype while mitigating the negative effects.

In the present study, we performed global identification of collar region-preferential genes (CRPGs) by integrating our RNA sequencing (RNA-Seq) data with various anatomical transcriptome datasets. Subsequently, 657 CRPGs were identified, and after validating the collar-preferential expression pattern through qRT-PCR, functional enrichment analysis using Gene Ontology (GO) and MapMan toolkit was conducted. Five regulatory genes identified from the transcription term in GO were suggested as downstream TFs of *OsLG1*, a key regulatory gene in the initiation of collar development in rice. Based on these findings, a conceptual regulatory model for collar development was proposed. Our findings could accelerate the modification of compacted leaf structure in rice to improve productivity while efficiently using energy, aligning with sustainable agriculture trends.

## 2. Results

### 2.1. Genome-Wide Identification of Collar Region-Preferential Genes

To investigate CRPGs, we grouped tissues according to their properties. The collar and ligule were grouped as the collar region, whereas the leaf blade and sheath were grouped as the leaf region (Figure 1a). RNA samples from collar, ligule, and sheath were collected from 5- to 6-week-old rice plants in the vegetative stage, and the expression raw data of leaf, shoot, root, anther, pollen, seed, and callus were collected from NCBI Sequence Read Archive (SRA; http://ncbi.nlm.nih.gov/sra/, accessed on 28 December 2022). Further, they were normalized using a previously described method all at once [28]. By comparisons of collar and leaf regions, we preliminary identified 4554 differentially upregulated genes in the collar region, with a log2 fold change > 1 and a *p*-value < 0.05. These genes were further classified into 12 distinct clusters using K-means clustering (KMC) analysis. Two of the twelve clusters showed collar region-preferential expression patterns. After manually filtering genes with a log2 intensity value >3 in the collar region, we finally identified 657 CRPGs (Figure 2). The expression heatmap of the identified CRPGs was reconstructed using MeV software (Figure 1b).

### 2.2. Authentication of Tissue-Preferential Expression Patterns of Six CRPGs

To verify the reliability of the collar region-preferred expression patterns of our candidate genes, we randomly selected six genes and performed in silico expression and quantitative real-time PCR analysis. The genes included one secondary metabolism related gene (*CRPG15*, *Os04g01810*), one TF (*CRPG32*, *Os11g02520*), one hormone-related gene (*CRPG33*, *Os09g16030*), one transferase (*CRPG70*, *Os10g04400*), one unannotated gene (*CRPG160*, *Os03g41480*), and one protein kinase (*CRPG348*, *Os04g15630*). Anatomical expression data were downloaded from the CAFRI-Rice database [28] by entering locus IDs retrieved from the Rice Genome Annotation Project website (RGAP; http://rice.plantbiology.msu.edu/, accessed on 28 December 2022). The data were reconstructed using MeV software. Real-time PCR experiments confirmed the expression of the six genes (Figure 3). As expected, all these showed preferential expression patterns in the collar tissue, corroborating our previous identification results.

### 2.3. Literature Analysis of Functionally Characterized CRPGs

Detailed archives of functionally characterized genes are useful for interpreting the functional significance of genes identified from large-scale datasets, and they provide a valuable resource that may facilitate further application. We used two databases, funRiceGenes (http://funricegenes.github.io/, accessed on 28 December 2022) [29] and Overview of Functionally Characterized Genes in Rice Online (OGRO; http://qtaro.abr.affrc.go.jp/ogro/table, accessed on 28 December 2022) [30] to identify 31 characterized genes among 657 CRPGs (Table 1; Appendix A). In terms of collar region development and LA formation, three CRPGs have been reported previously. Mutations in *OsLG1* lead to complete loss of collar, ligule, and auricle tissues, resulting in the erect leaf phenotype [25]. A bHLH transcription factor, namely *OsBC1*, regulates LA development by forming a trimeric complex [27], whereas its interaction partner, *OsBUL1*, was excluded from analysis in this study owing to its ubiquitous expression. Epigenetic modification of *RAV6* affects LA and grain size in a brassinosteroid-dependent manner [31]. Additionally, *OsGA2ox6* [32], *OsMPH1* [33], and *RePRP2.1* [34] alter cell elongation in other tissues, hinting the possibility that their parallel function in the collar region is to be expected. Further, more than half of the characterized CRPGs were phytohormone-related or stress-responsive genes, emphasizing the intricate internal signaling networks coupled with the development and regulation of collar region plasticity in rice.

### 2.4. Functional Enrichment Analysis and Classification of CRPGs

To explore the biological role of the candidate genes, we conducted GO enrichment analysis using the Rice Oligonucleotide Array Database (ROAD; http://www.ricearray.org/, accessed on 28 December 2022) [66]. Based on the criteria query number >2, hypergeometric *p*-value <0.05, and fold-enrichment value >2, we found that 15 GO terms were over-represented in CRPGs. These included five terms associated with lipid metabolism: oxylipin biosynthetic process (13.69; GO:0031408); fatty acid biosynthetic process (5.02; GO:0006633); lipid biosynthetic process (4.66; GO:0008610); steroid biosynthetic process (4.50; GO:0006694); and lipid metabolic process (3.05; GO:0006629). In addition, seven terms were associated with response and regulation: response to stimulus (9.02; GO:0050896); regulation of nitrogen utilization (7.76; GO:0006808); negative regulation of catalytic activity (4.08; GO:0043086); transcription (2.86; GO:0006350); regulation of transcription (2.73; GO:0045449); response to stress (2.66; GO:0006950); and apoptosis (2.19; GO:0006915). Also, three other terms were enriched: auxin-mediated signaling pathway (4.35; GO:0009734); metal ion transport (4.21; GO:0030001); and protein ubiquitination (3.43; GO:0016567) (Figure 4).

Terms for oxylipin biosynthetic process were the most enriched, as represented by lipoxygenase (LOX) and the specialized cytochrome P450 enzyme, allene oxide synthase (AOS). Their expression is consistent with the elevated expression of rice jasmonic acid (JA) biosynthetic genes, such as *OsLOX2* and *OsAOS1*, in the collar compared with the adjacent leaf region [10]. The only enriched pathway in our KEGG analysis was alpha-linolenic acid metabolism (Appendix A). Alpha-linolenic acid serves as the substrate for LOX in the biosynthesis of JA, which is the most extensively studied oxylipin in plants [67]. The subsequent process is taken over by AOS [68]. This finding was further supported by the regulation overview visualized through MapMan analysis (Appendix A). The aforementioned genes were classified into functional groups associated with jasmonate synthesis, indicating the active biosynthesis of JA and its crucial biological functions in the collar region. In addition, the identification of auxin-related genes in the MapMan overview analysis was consistent with the terms “response to stimulus” and “auxin-mediated signaling pathway” in the GO analysis results. Other hormone-related genes were also observed in the hormone metabolism category. Additionally, approximately half of the total mapped CRPGs were associated with transcription (GO:0006350) and regulation of transcription (GO:0045449), highlighting the importance of TFs in tissue-preferential regulation. Overall, our functional analysis revealed the tissue-preferential regulation by TFs and phytohormones, particularly JA and auxin, and emphasized their intricate networks in relation to collar region development and LA formation.

### 2.5. Case Study Using the lg1 Mutant Revealed Downstream Regulatory Elements

Among the functionally characterized CRPGs, *LIGULELESS1* (*LG1*) is a key gene that initiates collar region organogenesis. Knockout of the *LG1* gene leads to the entire loss of the collar region and an erect leaf structure in rice. We identified a T-DNA insertion line (3A-12312) of *OsLG1/LOC_Os04g56170* in our T-DNA mutant pool (Figure 5e). T-DNA was inserted into the first exon of *OsLG1*, and wild-type plants and *oslg1* mutants were segregated through genotyping experiments (Figure 5f). Quantitative real-time PCR analysis showed that there was no *OsLG1* mRNA detected in 3A-12312 line homozygous plants (Figure 5g). As previously reported, *oslg1* plants exhibited a complete absence of the collar region, and 4-week-old *lg1* plants showed a more obvious erect leaf structure than wild-type plants (Figure 5a–d). *LG1* is a member of the SQUAMOSA PROMOTER BINDING-LIKE (SPL) TF family. In *Arabidopsis* and wheat, *LG1* orthologous genes directly bind to the promoter region containing the GTAC core motif [69,70] (Figure 5h). Considering the conserved function of *LG1* in multiple crops (rice, maize, and wheat) [70,71], we hypothesized that CRPGs containing GTAC motifs in their promoters could potentially be regulated by *LG1*. To identify potential targets, we used Find Individual Motif Occurrences (FIMO), which is embedded in the MEME tool, to screen for TFs among CRPGs that possess the GTAC motif within their 2 kb promoter region. As a result, we identified five TFs as potential downstream regulatory genes of *OsLG1*. The expression levels of the genes encoding these five TFs were significantly reduced in the *oslg1* mutant compared with wild-type plants, supporting our hypothesis regarding the regulatory pathway (Figure 5i–m). However, the direct binding ability of *OsLG1* to the GTAC motif of the five CRPGs requires further investigation.

## 3. Discussion

Manipulating LA is an important trait for the efficient use of limited land and energy resources. Due to its importance, several studies have illuminated the organogenesis process of the collar region, which is a determinant organ of LA, and the involving genetic factors in different perspectives [10,11]. Despite the collar region mRNA profiling in prior studies, it is necessary to identify additional candidate genes with organ-preferential expression owing to the developmental and positional dynamics of the collar region. Also, previous studies have predominantly focused on stage-specific genes during collar region development, but the pleiotropic expression of these genes poses a major challenge for practical applications. To overcome this challenge and offer a more comprehensive set of potential candidates, we performed genome-wide identification and functional analysis of CRPGs in rice.

Through the integration of our RNA-Seq analysis with NCBI SRA datasets, we successfully identified 657 CRPGs. To validate the reliability of our in silico expression analysis data, we conducted expression testing on a randomly selected subset of six CRPGs (Figure 1, Figure 2 and Figure 3). Functional enrichment analyses using GO and KEGG provided valuable insights into the biological implications of these genes (Figure 4). Notably, the most prominent terms among genes exhibiting preferential expression in the collar region were phytohormone-related, particularly JA biosynthesis and auxin signaling. Previous studies have shown that auxin has a negative effect on collar region development and regulation, including biosynthesis and signaling, while also interacting with other phytohormones or nutrients [72]. Given the complexity of this regulatory system, it may be more effective to maximize the responsiveness of tissue-specific signaling genes rather than fine-tuning hormone levels to control leaf inclination. Regarding JA, except for MeJA, which represses leaf inclination through a brassinosteroid-dependent mechanism [73], there is limited information on the action of JA-related genes, specifically in the collar region, while the involvement of JA in collar region development has been hinted. In line with this notion, although no previously functionally characterized genes related to programmed cell death (PCD) have been identified, the term “apoptosis” was enriched in our study. Notably, Zhou et al. (2017) [10] highlighted the crucial role of PCD in the overall development of the collar region, yet PCD-related terms were not specifically identified in their study. This discrepancy may be attributed to the unspecified expression of the genes analyzed. In summary, the functional assessment of CRPGs in relation to these relevant terms may provide insights into the molecular mechanisms underlying JA or PCD in the collar region that are yet to be elucidated.

The *LG1* gene (Table 1) is renowned for its involvement in the erect leaf phenotype in crops, making it a promising target for crop improvement [26]. However, the research on the downstream mechanisms of *LG1* has not advanced proportionally to its significance. In our study, we identified five downstream regulatory genes of *OsLG1* among the CRPGs (one each from the MYB, NAC, bZIP, bHLH, and Dof TF families) (Figure 5). Previous studies have documented the role of these TF families in governing LA regulation. For example, *OsMYB7* contributes to leaf inclination by modulating auxin levels and promoting cell elongation on the adaxial side [74]. Although the erect leaf phenotype offers advantages in terms of enhanced yield under certain conditions, for the adaptation of crops to diverse geographical environments, it is considered more desirable to have erect leaves while retaining some degree of plasticity rather than a complete loss of structural flexibility. From this perspective, the five CRPGs downstream of *OsLG1* hold potential as promising candidates for exhibiting the desired phenotype without negatively impacting plant growth. Although further investigation into the underlying molecular mechanisms is necessary, we present an overall hypothetical model that will be valuable for future studies (Figure 6).

Direct-sowing cultivation has garnered considerable attention as a strategy to reduce carbon emissions. However, direct-sowing often involves dense planting, which presents challenges in terms of energy utilization efficiency and disease management [75]. As a consequence, the stress-related CRPGs identified in our study, along with the aforementioned candidates, hold potential not only for uncovering molecular mechanisms but also for practical applications. Moreover, the development of the collar region exhibits a certain level of conservation across various crops, providing a valuable foundation for expanding research to other crops that have received comparatively less attention than rice. Collectively, comprehensive investigations and dissections of currently identified CRPGs will be the cornerstone of biotechnological applications and crop breeding to achieve the so-called geo-adapted crops with optimal plant ideotype.

## 4. Materials and Methods

### 4.1. Plant Materials and Growth Conditions

Dongjin rice (*O. sativa* ssp. *japonica*) and T-DNA insertional line (3A-12312, cv. Dongjin) seeds were respectively sterilized with 50% sodium hypochlorite and germinated on half-strength Murashige and Skoog medium for one week in an incubator at 28 °C/22 °C (day/night) under continuous light conditions. Seedling plants (7 days old) were transferred to the soil condition in a greenhouse or artificial growth chamber (28 °C/25 °C day/night, 14/10 h light/dark, and 80% relative humidity) at the Kyung Hee University, Yongin, Republic of Korea, for further development.

### 4.2. RNA-Seq Analysis and Data Collection

The collar, ligule, and sheath from 5- to 6-week-old WT vegetative-stage plants (cv. Dongjin) cultivated under greenhouse conditions were sampled for RNA-Seq analysis. We used a RNeasy Plant Mini Kit from Qiagen to extract RNA from samples and then a TruSeq Stranded mRNA LT Sample Prep Kit for library construction. The libraries were sequenced on an Illumina platform (Illumina NovaSeq 6000) [76], with two biological replicates for each sample. Raw expression datasets for leaf, shoot, root, anther, pollen, seed, and callus tissues were downloaded from NCBI SRA (http://ncbi.nlm.nih.gov/sra/, accessed on 28 December 2022). An identical pipeline was used for preprocessing and reference genome alignment of the sequenced raw data of the abovementioned tissues, followed by normalization, as described previously [28]. The accession number for the collar, ligule, and sheath is E-MTAB-11005 in ArrayExpress; that for leaf, shoot, root, anther, pollen, seed, and callus tissues is DRP000391 in NCBI SRA.

### 4.3. Identification of Candidates Coupled with Heatmap Analysis

Using the DESeq2 package, we conducted statistical testing based on normalized read counts to identify differentially expressed genes (DEGs) upregulated in the collar region compared with the leaf region. We selected genes with a log2 fold change >1 and a *p*-value < 0.05. The integrated log2 intensity values of the resulting upregulated DEGs were then loaded into the MeV program (version 4.9.0), where KMC analysis was performed using the Euclidean distance algorithm [77]. This clustering analysis grouped the upregulated DEGs into 12 distinct clusters, from which two that exhibited a preferential expression pattern in the collar region were selected. Subsequently, we applied an additional filter requiring a log2 intensity value >3 in the collar region. Finally, a heatmap showing the expression patterns of these final candidates was generated using the MeV program (version 4.9.0) and employing the single-color array method. Information regarding the 657 CRPGs identified is listed in Appendix A.

### 4.4. Literature Analysis

The OGRO (http://qtaro.abr.affrc.go.jp/ogro/table, accessed on 28 December 2022) and funRiceGenes (http://funricegenes.github.io/, accessed on 28 December 2022) databases were searched to identify functionally characterized CRPGs. Detailed information regarding these functionally characterized CRPGs is provided in Table 1.

### 4.5. Enrichment Analysis Via GO, KEGG, and MapMan

The Gene IDs of the 657 CRPGs were used as entries for query mapping in ROAD (http://ricephylogenomics-khu.org/road/home.php, accessed on 28 December 2022). To identify significant GO terms (GO type: biological process), certain criteria were applied: query number >2; fold-enrichment value >2; and hypergeometric *p*-value < 0.05 [78]. The fold-enrichment values were obtained by dividing the query number by the query-expected value. To conduct KEGG enrichment analysis, we used R studio (2023.06.0+421) along with the clusterProfiler package version 4.8.1. The input data included cluster information and Rice Annotation Project Database IDs (http://rapdb.dna.affrc.go.jp, accessed on 28 December 2022), with the organism code for rice specified as “dosa”. The results were filtered by employing an adjusted *p*-value threshold of <0.05 [79]. For visualization, R studio version 4.3.0 and the ggplot2 package version 3.4.2 were used [80]. MapMan version 3.6.0RC1 was used for visualizing CRPGs mapped to various pathways or processes [81]. An overview of metabolism and regulation was analyzed. Detailed information regarding GO and MapMan analysis is presented in Appendix A and Appendix A, respectively.

### 4.6. RNA Extraction and qRT-PCR

Wild-type plants (cv. Dongjin) and *lg1* mutants were grown under growth chamber conditions prior to RNA extraction. Subsequently, 1 cm of the collar region was collected and immediately frozen in liquid nitrogen. Total RNA was isolated manually (RNAiso; Takara Bio, Shiga, Japan) and quantified using a NanoDrop Spectrophotometer ND-2000 [82]. After synthesizing cDNAs using SuPrimeScript RT Premix [with oligo (dT), 2×] (GeNet Bio, Daegu, Republic of Korea), qRT-PCR was performed on a Rotor Gene Q instrument system (Qiagen, Hidden, Germany) using SYBR Green I (GeNet Bio, Republic of Korea). All reactions were conducted in three biological replications, and data analysis was performed using the 2^−ΔΔCt^ method, as previously described [83]. For other tissues, samples were collected as previously reported [84]. All the primers used in this study are listed in Appendix A.

### 4.7. Motif Scanning

The motif matrix profile of the GTAC core motif (MA0578.1) was downloaded from JASPAR 2022 in MEME format [85]. The function FIMO (Find Individual Motif Occurrences) in MEME suite (version 5.5.3) was used for scanning the 2 kb upstream promoter sequence of TFs [86] among CRPGs that matched with the GTAC motif [87]. The resulting information is in Appendix A.

## Figures and Tables

**Figure 1 plants-12-02959-f001:**
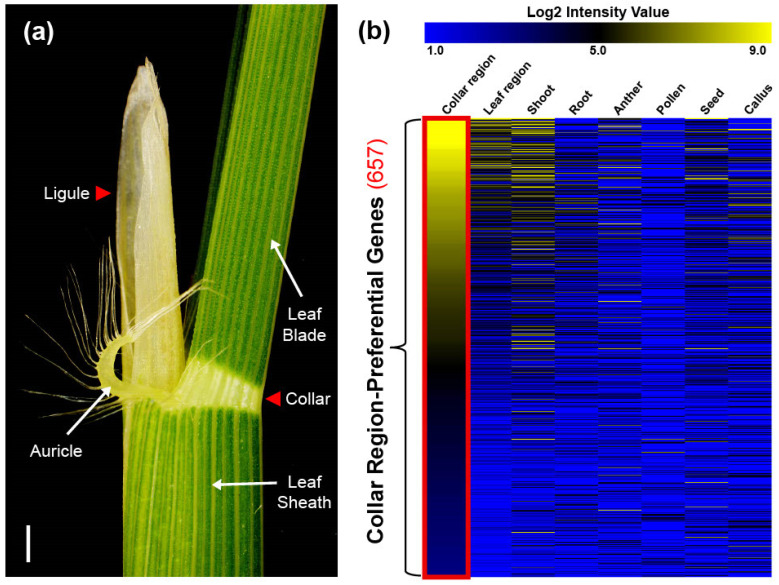
Illustration of the collar region and heatmap showing the expression of 657 collar region-preferential genes (CRPGs). (**a**) The collar region adjacent part in 5-week-old rice plants (cv. Dongjin). The collar region is marked by a red triangle. Scale bar = 0.5 cm. (**b**) Expression of CRPGs in eight anatomical tissues, including the collar region, leaf region, shoot, root, anther, pollen, seed, and callus. The color scheme ranging from blue to black to yellow represents the strength of normalized log2 intensity values. The red rectangle highlights collar region-preferential expression.

**Figure 2 plants-12-02959-f002:**
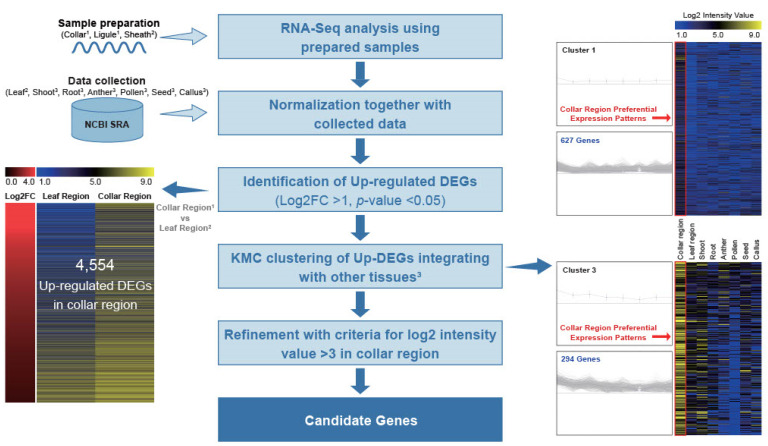
Workflow for the genome-wide identification of CRPGs. Light and deep blue horizontal arrows represent input data and output results, respectively. Upregulated differentially expressed genes (DEGs) were identified by comparing the collar region with the leaf region. In the left panel, the output shows the log2 fold change combined with anatomical expression data of upregulated DEGs (collar region vs. leaf region), where the red color indicates upregulation in the collar region. The color scheme ranging from blue to black to yellow represents the strength of normalized log2 intensity values. The right panel displays a centroid graph with expression images of collar region-preferential gene clusters obtained from KMC analysis. The number of genes in each cluster is shown, and expression in the collar region is highlighted with a red rectangle. Collar region^1^: collar^1^ and ligule^1^; leaf region^2^: leaf^2^ and sheath^2^; other tissues^3^: shoot^3^, root^3^, anther^3^, pollen^3^, seed^3^, and callus^3^.

**Figure 3 plants-12-02959-f003:**
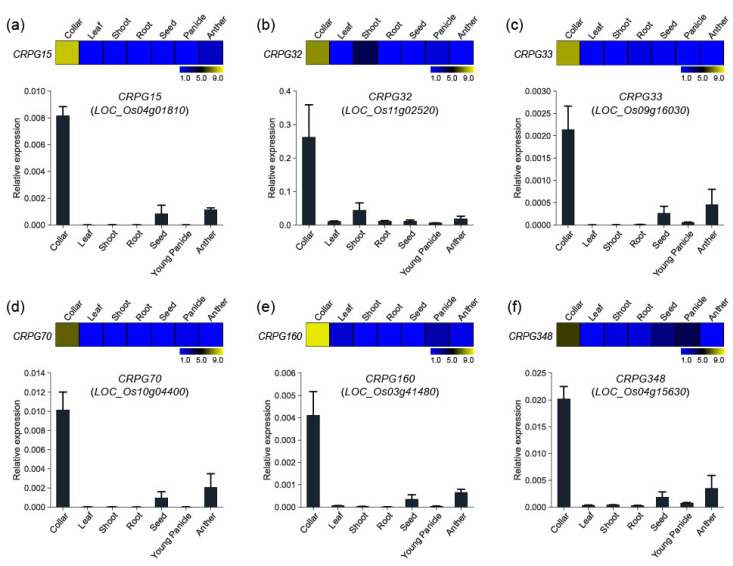
Expression validation of six CRPGs. (**a**) *CRPG15*/*Os04g01810*, (**b**) *CRPG32*/*Os11g02520*, (**c**) *CRPG33*/*Os09g16030*, (**d**) *CRPG70*/*Os10g04400*, (**e**) *CRPG160*/*Os03g41480*, and (**f**) *CRPG348*/*Os04g15630*. Expression heatmaps were constructed using the MeV program, and numeric values represent the average normalized log_2_ intensity values obtained from the CAFRI-Rice database. Collar region-preferential expression pattern was examined using real-time PCR analysis in seven distinct tissues (collar, leaf, shoot, root, seed, young panicle, and anther). *Ubi5* (*Os01g22490*) was used as an internal control in real-time PCR experiments.

**Figure 4 plants-12-02959-f004:**
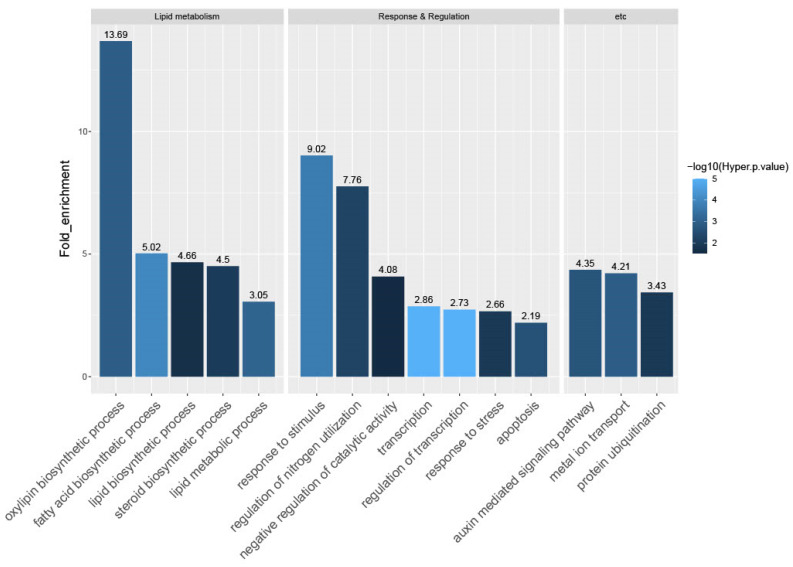
Gene Ontology (GO) enrichment analysis of CRPGs. The 657 CRPGs were mapped to GO terms in the biological process category. Enriched GO terms were categorized into three groups, including lipid metabolism, response and regulation, and others. The numbers above the bar represent the fold-enrichment value of the GO terms. The colors of the bars indicate statistical significance [−log10(hyper *p*-value)].

**Figure 5 plants-12-02959-f005:**
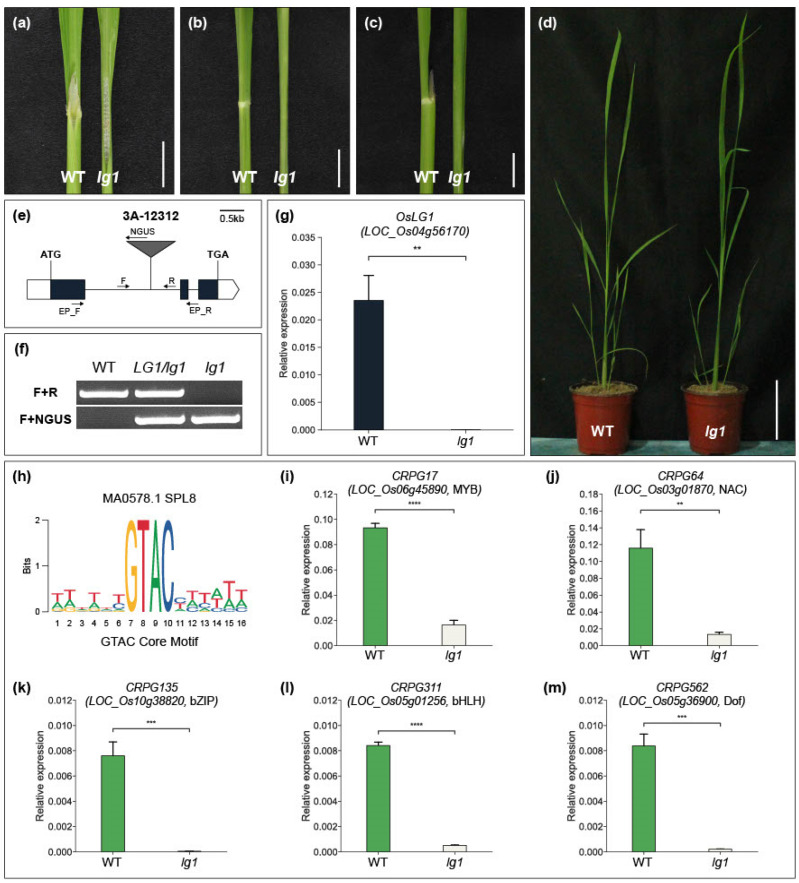
Phenotype of the rice *lg1* mutant and discovery of downstream regulatory factors of *OsLG1*. (**a**–**d**) Comparison of wild-type (WT) plant (**left**) and *lg1* mutant (**right**) collar region, showing the adaxial side (**a**), abaxial side (**b**), lateral side (**c**), and overall features (**d**). Scale bars in (**a**–**c**) indicate 1 cm, and in (**d**) indicate 10 cm. (**e**) Schematic diagram of T-DNA insertional line 3A-12312 for *OsLG1*. White boxes represent the 5′ and 3′ UTR regions, and black boxes represent the exon region. T-DNA was inserted in the first intron. F, R, and NGUS are primers used for genotyping. EP_F and EP_R are primers for checking expressions. Scale bar: 0.5 kb. (**f**) Segregation analysis of the *lg1* mutant through genotyping. (**g**) Expression validation of *OsLG1* in WT and *lg1* mutant. (**h**) Sequence logo for GTAC core motif. (**i**–**m**) Expression of five candidate genes in the WT and *lg1* mutant. The internal control used in this study was *OsUbi5* (*Os01g22490*). The experiment included three biological replicates, and the *t*-test was conducted on independent samples with Bonferroni correction. ** 0.001 < *p* ≤ 0.01; *** 0.0001 < *p* ≤ 0.001; **** *p* ≤ 0.0001.

**Figure 6 plants-12-02959-f006:**
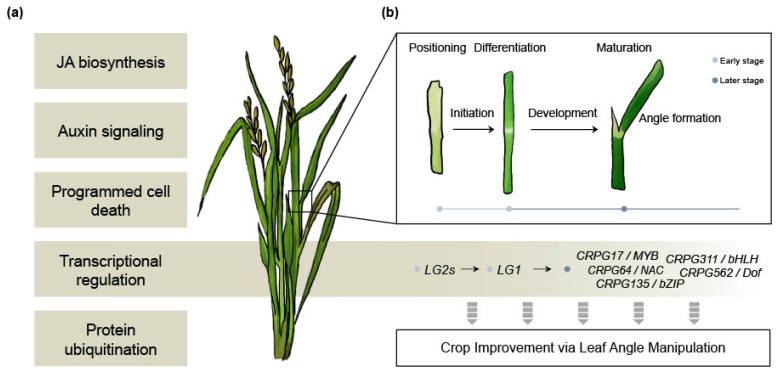
Overall hypothetical model. (**a**) Biological processes involved in collar region development. (**b**) Detailed developmental processes in the sequence of leaf angle formation, along with potential molecular regulators that could serve as candidates for crop improvement.

**Table 1 plants-12-02959-t001:** Summary of functionally characterized genes among 657 CRPGs in rice.

Locus	Symbol	Category	Keyword	References
LOC_Os01g43650	OsWRKY11	R, T	Heat and drought tolerance, Disease resistance	[35,36]
LOC_Os01g48290	OsDof4	P	Flowering time	[37]
LOC_Os02g02930	OsLIS	R, T	Blast resistance, JA	[38]
LOC_Os02g12350	HDA703	M	Panicle development, Fertility	[39]
LOC_Os02g12680	OsAOS3|OsHPL2	R, T	Blight resistance	[40]
LOC_Os02g26430	OsWRKY42	P, R, T	Leaf senescence, Blast susceptibility, JA	[41,42]
LOC_Os02g35970	CPT1	M	Root phototropism, Auxin	[43]
LOC_Os02g41954	GA2ox9	M	Dwarfism, GA	[44]
LOC_Os02g43330	OsHOX24|OsSLI1	R, T	Abiotic stress response, ABA	[45,46]
LOC_Os02g45850	RAV6	M	Leaf angle, Seed size, BR	[31]
LOC_Os03g12500	OsAOS2	R, T	Defense response, JA	[47]
LOC_Os03g44710	OsSh1	M	Seed shattering	[48]
LOC_Os04g01950	BLEC-Str8	R, T	Salt stress	[49]
LOC_Os04g20330	OscZOG1	M	Grain-yielding traits, Cytokinin	[50]
LOC_Os04g44150	OsGA2ox6	M	Dwarf, GA	[32]
LOC_Os04g48350	OsDREB1E	R, T	Drought tolerance	[51]
LOC_Os04g52310	OsZIP3	P	Zn distribution	[52]
LOC_Os04g56170	OsLG1	M	Liguleless, Closed panicle	[25,53]
LOC_Os05g01140	OsJMT1	R, T	Defense response, JA	[54]
LOC_Os06g03670	OsDREB1C	M, R, T	Abiotic stress tolerance, Growth retardation	[55]
LOC_Os06g45890	OsMPH1|OsMYB45	M, R, T	Grain yield, Plant height, Cadmium tolerance	[33,56]
LOC_Os07g05940	OsNCED4	R, T	Drought tolerance, ABA	[57]
LOC_Os07g23660	RePRP2.1	M	Root cell elongation, ABA	[34]
LOC_Os09g26999	OsDEP1|DN1|qPE9-1	M	Erect panicle, Dwarf, Grain yield, GA	[58,59]
LOC_Os09g28440	OsEATB	M	Dwarf, Tillering, Panicle branching, Ethylene, GA	[60]
LOC_Os09g33580	OsBC1	M	Grain size, Leaf angle	[27]
LOC_Os09g37400	OsSAUR45	M	Plant growth, Auxin	[61]
LOC_Os10g39260	OsAP77	R, T	Defense response, ABA, SA	[62]
LOC_Os11g02520	OsWRKY89	R, T	Abiotic and biotic stress, JA	[63]
LOC_Os11g29840	LA1|LAZY1	M	Tiller angle, Auxin	[64]
LOC_Os12g03050	ONAC300	M	Shoot apical meristem	[65]

Category: R, resistance; T, tolerance; M, morphological trait; P, physiological trait.

## Data Availability

The expression data presented in this study are available in the Appendix A.

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
