# Peer review of "Genome-Wide Identification and Analysis of Collar Region-Preferential Genes in Rice"

_plants, 2023, doi:10.3390/plants12162959_

Round 1
Reviewer 1 Report
The paper titled "Genome-wide Identification and Analysis of Collar Region-Preferential Genes in Rice" presents a thorough investigation into collar region-preferential genes in rice, utilizing genome-wide identification and analysis methods. The research topic holds great relevance, as it delves into the crucial understanding of gene regulation in specific genome regions for crop improvement. The introduction offers a clear rationale for the study, highlighting existing knowledge gaps, and establishing a strong foundation for the subsequent sections.
To further enhance the quality of the article, the authors can incorporate additional bioinformatics analyses, such as gene structure analysis, exon-intron analysis, chromosome mapping, phylogenetic analysis, and protein-protein interaction. These comprehensive analyses would provide readers with a more detailed understanding of collar region-preferential genes in rice.
The methods section is well-crafted and sufficiently detailed, enabling readers to grasp the experimental design and data analysis. The study's rigor is demonstrated through the appropriate use of methodologies, including data mining, bioinformatics tools, and statistical analyses. To promote reproducibility and facilitate further research, it is advisable for the authors to provide specific information on the software used, version numbers, and parameter settings.
Furthermore, to enrich the discussion, the authors should consider incorporating more comparative analyses involving other plant species. Such comparisons would lead to a more comprehensive comprehension of collar region-preferential gene conservation and evolution.
The conclusion effectively summarizes the study's main findings, reaffirming the significance of collar region-preferential genes in rice and their potential agricultural applications. The conclusions drawn are well-supported by the data presented in the results section. Nonetheless, including a clear outline of future research directions would be beneficial, as it would inspire further investigations in this field.
Author Response
Comments 1:
To further enhance the quality of the article, the authors can incorporate additional bioinformatics analyses, such as gene structure analysis, exon-intron analysis, chromosome mapping, phylogenetic analysis, and protein-protein interaction. These comprehensive analyses would provide readers with a more detailed understanding of collar region-preferential genes in rice.
Response to comment 1:
We appreciate your careful comment. The purpose of this study is to uncover potential candidates for crop improvement by leaf angle manipulation. Hence, we identified 657 CRPGs using multiple omics data and revealed the biological meanings of these candidate genes via functional enrichment analyses. Further, mutant analysis suggested the molecular mechanisms related to the regulation of leaf structure. Regarding the analyses suggested, we believe that further study is needed. For example, selecting a CRPG from the enriched GO terms and performing analyses (mentioned above) for its gene family in the rice genome will help to understand the specialized function of the gene in the collar region. Also, if possible, comparing the gene family among different species will give an insight into the evolutional conservation and dynamics. In summary, the comments suggested are mostly related to a gene family study of interest, but our study identified a global candidate genes showing collar region preferred expression which have diverse family genes and genes not beloning gene families. If you excuse us, your suggestion is likely to have more value through follow-up studies.
Comments 2:
The methods section is well-crafted and sufficiently detailed, enabling readers to grasp the experimental design and data analysis. The study's rigor is demonstrated through the appropriate use of methodologies, including data mining, bioinformatics tools, and statistical analyses. To promote reproducibility and facilitate further research, it is advisable for the authors to provide specific information on the software used, version numbers, and parameter settings.
Response to comment 2:
To address this comment, we added version information about the MeV program (lines 329, 335), two databases used for literature analysis (lines 338-339), and R studio (line 348).
Comments 3:
Furthermore, to enrich the discussion, the authors should consider incorporating more comparative analyses involving other plant species. Such comparisons would lead to a more comprehensive comprehension of collar region-preferential gene conservation and evolution.
Response to comment 3:
We sincerely thank the valuable comment. We have checked the literature carefully and found that most of the research in this area has focused on crops1 (mainly rice and maize), and despite the early identification of key genes (LGs) involved in maize collar region2-3 (ligule in maize), the following research focused more on rice4, especially on genome-wide study5. In other plant species except rice, we found no reports related to the global identification of collar region-preferential genes. Thus, we expect that the current study could fill the gap and will become a reference in comparative studies of collar region-preferential genes among crops.
Reference
- Cao Y, Zhong Z, Wang H, Shen R. Leaf angle: a target of genetic improvement in cereal crops tailored for high-density planting. Plant Biotechnol J. 2022 Mar; 20(3): 426-436. doi: 10.1111/pbi.13780.
- Moreno MA, Harper LC, Krueger RW, Dellaporta SL, Freeling M. liguleless1 encodes a nuclear-localized protein required for induction of ligules and auricles during maize leaf organogenesis. Genes Dev. 1997 Mar 1; 11(5): 616-28. doi: 10.1101/gad.11.5.616.
- Walsh J, Waters CA, Freeling M. The maize gene liguleless2 encodes a basic leucine zipper protein involved in the establishment of the leaf blade-sheath boundary. Genes Dev. 1998 Jan 15; 12(2): 208-18. doi: 10.1101/gad.12.2.208.
- Xu J, Wang JJ, Xue HW, Zhang GH. Leaf direction: Lamina joint development and environmental responses. Plant Cell Environ. 2021 Aug; 44(8): 2441-2454. doi: 10.1111/pce.14065.
- Zhou LJ, Xiao LT, Xue HW. Dynamic Cytology and Transcriptional Regulation of Rice Lamina Joint Development. Plant Physiol. 2017 Jul; 174(3): 1728-1746. doi: 10.1104/pp.17.00413.
Comments 4:
The conclusion effectively summarizes the study's main findings, reaffirming the significance of collar region-preferential genes in rice and their potential agricultural applications. The conclusions drawn are well-supported by the data presented in the results section. Nonetheless, including a clear outline of future research directions would be beneficial, as it would inspire further investigations in this field.
Response to comment 4:
Thank you for your comment. We have provided a systemic view of collar region-preferential genes in this study; however, further functional studies of individual genes would be interesting to work on, particularly for genes belonging to highlighted processes (lines 267-270, 287-292). Also, comparative analysis will help to understand the conservation and divergence of CRPGs in different plant species (lines 292-294). Further, we have enriched the content according to the suggestion from reviewer #1 (lines 295-297).
Reviewer 2 Report
The manuscript by Jiang et al. reports genome wide identification and analysis of collar region preferential genes (CRPGs) in rice. They integrated the RNA-seq analysis with existing anatomical transcriptome datasets and identified hundreds of potential CRPGs. Several genes were selected for qRT-PCR verification and all of them showed collar preferential expression pattern. The authors then performed functional enrichment analysis as well as case study using the lg1 mutant, from which they identified five transcription factors as potential downstream regulatory genes of OsLG1.
The experiments in this manuscript are well designed and described clearly. The findings are novel and provide a great resource not only for molecular mechanisms investigation but also for practical applications in rice as well as other crops.
Author Response
We appreciate your effort to review our manuscript. You gives an accurate summary of our work and positive feedback. Thanks again for taking time to review this manuscript.
Reviewer 3 Report
1.“The collar, ligule, and sheath from 5 to 6-week-old WT vegetative-stage plants (cv. Dongjin) cultivated under greenhouse conditions were sampled for RNA-Seq analysis”
Fig1 . (b) “Expression of CRPGs 102 in eight anatomical tissues, including the collar region, leaf region, shoot, root, anther, pollen, seed, 103 and callus. “Leaf region is sheath or collected from NCBI Sequence Read Archive?
2. Some of collar region-preferential genes (CRPGs) are high expression in other tissues.

Author Response
Comments 1:
1.“The collar, ligule, and sheath from 5 to 6-week-old WT vegetative-stage plants (cv. Dongjin) cultivated under greenhouse conditions were sampled for RNA-Seq analysis”
Fig1 . (b) “Expression of CRPGs 102 in eight anatomical tissues, including the collar region, leaf region, shoot, root, anther, pollen, seed, 103 and callus. “Leaf region is sheath or collected from NCBI Sequence Read Archive?
Response to comment 1:
We thank the reviewer for this comment, leaf blade and leaf sheath together defined as leaf region in our study (lines 85-87, lines 114-115). The leaf sheath was sampled for RNA-seq analysis and normalized together with the expression raw data of leaf blade (lines 87-91). The accession number was given in lines 318-320, E-MTAB-11005 for the collar, ligule, and sheath; and DRP000391 for leaf, shoot, root, anther, pollen, seed, and callus.
Comments 2:
- Some of collar region-preferential genes (CRPGs) are high expression in other tissues.
Response to comment 2:
We deeply appreciate the reviewer’s suggestion. This comment could be explained in two aspects. First, the terminology “collar region-preferential genes” represents genes with relatively higher expression in the collar region than in other tissues. It differs from the term “collar region-specific genes” which means genes only expressed in the collar region. Thus, some CRPGs may have expression in other tissues. Second, the transcriptomic data provided by the RiceGE database do not cover the collar region, so it is hard to compare the relative expression of the collar region with other tissues. Also, the microarray techniques, which is well known that have many limitations than RNA-Seq such as low sensitivity and background noise, are used to generate transcriptomic data provided by the RiceGE database, and the scale of expression is quite wide in the RiceGE database (from 0.0 to nearly 5000.0 for the suggested genes). In our study, we used RNA-Seq for generating expression data and applied log2intensity value to narrow down the range of expression read counts which is suitable for distinguishing relative expression of genes. The collar region-preferential expression was confirmed by qRT-PCR experiments (Figure 3a, c, and f). Additionally, we confirmed collar region-preferential expression of two genes that were suggested by reviewer #3 (Figures in the attachment)

Round 2
Reviewer 1 Report
Thank you for taking into consideration the comments proposed to the authors.
Author Response
Comment 1: Thank you for taking into consideration the comments proposed to the authors.
Response: Due to the comment, our manuscript was more improved.